# A recurrent point mutation in *PRKCA* is a hallmark of chordoid gliomas

Shai Rosenberg [1,2], Iva Simeonova[1], Franck Bielle[1,3], Maite Verreault[1], Bertille Bance[1], Isabelle Le Roux[1], Mailys Daniau[1], Arun Nadaradjane[1], Vincent Gleize[1], Sophie Paris[1], Yannick Marie[1,4], Marine Giry[1], Marc Polivka[5], Dominique Figarella-Branger[6], Marie-Hélène Aubriot-Lorton[7], Chiara Villa[8], Alexandre Vasiljevic[9], Emmanuèle Lechapt-Zalcman[10], Michel Kalamarides[1,11], Ariane Sharif[12], Karima Mokhtari[1,3], Stefano Maria Pagnotta[13,14], Antonio Iavarone[14,15], Anna Lasorella[14,16], Emmanuelle Huillard[1] & Marc Sanson[1,4,17,18]

Chordoid glioma (ChG) is a characteristic, slow growing, and well-circumscribed diencephalic tumor, whose mutational landscape is unknown. Here we report the analysis of 16 ChG by whole-exome and RNA-sequencing. We found that 15 ChG harbor the same $PRKCA^{D463H}$ mutation. *PRKCA* encodes the Protein kinase C (PKC) isozyme alpha (PKCα) and is mutated in a wide range of human cancers. However the hot spot $PRKCA^{D463H}$ mutation was not described in other tumors. $PRKCA^{D463H}$ is strongly associated with the activation of protein translation initiation (EIF2) pathway. $PKCα^{D463H}$ mRNA levels are more abundant than wild-type PKCα transcripts, while $PKCα^{D463H}$ is less stable than the $PCKα^{WT}$ protein. Compared to $PCKα^{WT}$, the $PKCα^{D463H}$ protein is depleted from the cell membrane. The $PKCα^{D463H}$ mutant enhances proliferation of astrocytes and tanycytes, the cells of origin of ChG. In conclusion, our study identifies the hallmark mutation for chordoid gliomas and provides mechanistic insights on ChG oncogenesis.

[1] Inserm U 1127, CNRS UMR 7225, Sorbonne Universités, UPMC Univ Paris 06 UMR S 1127, ICM, F-75013 Paris, France. [2] Gaffin Center for Neuro-oncology, Sharett Institute for Oncology, Hadassah – Hebrew University Medical Center, 91120 Jerusalem, Israel. [3] Laboratoire R Escourolle, AP-HP, Hôpital de la Pitié-Salpêtrière, F-75013 Paris, France. [4] Onconeurotek Tumor Bank, Institut du Cerveau et de la Moelle épinère—ICM, F-75013 Paris, France. [5] Department of Pathology, AP-HP, Hôpital Lariboisière, F-75010 Paris, France. [6] Pathology and Neuropathology Department, Assistance Publique-Hôpitaux de Marseille (AP-HM), CHU Timone, 13005 Marseille, France. [7] Department of Pathology and CRB Ferdinand Cabanne, CHU Dijon Bourgogne, 21000 Dijon, France. [8] Department of Pathological Cytology and Anatomy, Foch Hospital, Suresnes F-92151 Paris, France. [9] Centre de Biologie et Pathologie Est, Groupement Hospitalier Est, Hospices Civils de Lyon, 69500 Bron, France. [10] Department of Pathology, CHU de Caen, Caen, France Normandie Univ, UNICAEN, CEA, CNRS, ISTCT/LDM-TEP Group, 14000 Caen, France. [11] Service de Neurochirurgie, AP-HP, Hôpital de la Pitié-Salpêtrière, F-75013 Paris, France. [12] INSERM U1172, "Development and Plasticity of the Neuroendocrine Brain", F-59045 Lille, France. [13] Dipartimento di Scienze e Tecnologie, Università degli Studi del Sannio, 82100 Benevento, Italy. [14] Institute for Cancer Genetics, Columbia University Medical Center, New York City, NY 10032, USA. [15] Departments of Neurology and Pathology, Institute for Cancer Genetics, Irving Comprehensive Research Center, New York, NY 10032, USA. [16] Departments of Pediatrics and Pathology, Institute for Cancer Genetics, Irving Comprehensive Research Center, New York, NY 10032, USA. [17] AP-HP, Hôpital de la Pitié-Salpêtrière, Service de Neurologie 2, F-75013 Paris, France. [18] Site de Recherche Intégrée sur le Cancer (SiRIC) "CURAMUS", F-75013 Paris, France. Correspondence and requests for materials should be addressed to M.S. (email: marc.sanson@aphp.fr)

Chordoid glioma (ChG) is a rare but characteristic slow growing, well-circumscribed, WHO grade II brain tumor[1]. This tumor develops from the anterior part of the third ventricle[2] and occurs mostly in adults (mean age 47), with a female predominance[3]. ChG may be revealed by headache, visual deficits, memory impairment, or endocrine disturbance[2]. The MRI typically shows a hypothalamic well-defined mass with homogeneous contrast enhancement. Histological architecture—reminiscent of chordoma—consists of cords of glial cells surrounded by an abundant mucinous stroma widely infiltrated with lymphocytes and plasma cells[1]. Immunohistochemical, ultrastructural, and transcriptomic studies suggest that ChG derive from tanycytes, a specific type of ependymal cells from the circumventricular organs[4,5].

The underlying genetic alterations that cause ChG remain obscure. Major obstacles to molecular characterization of ChG have been the infrequent incidence of the disease and the scarcity of tumor tissues available for molecular analyses. Here, we collected 16 primary ChG tumors and analyzed them by next-generation sequencing (NGS) to identify potentially recurrent somatic alterations. These studies and the follow-up experimental validation identify a novel $PRKCA^{D463H}$ mutation, as the disease-defining genetic event of ChG.

## Results

### ChG are specifically mutated on $PRKCA^{D463H}$.

Sixteen patients with ChG (Fig. 1a) and available tumor sample were recruited: four frozen tumor tissues with matched normal blood DNA (G1 group), six frozen tumors without matched normal DNA (G2 group), and six formalin-fixed paraffin-embedded (FFPE) tumor samples without matched normal DNA (G3 group). G1 and G2 samples were processed for whole-exome sequencing (WES) and RNA-sequencing (RNA-Seq). WES of genomic DNA from the four G1 tumors identified a total of 346 somatic mutations with a median of 73 per tumor (range 17–184). All four tumors harbored a mutation mapping on chromosome 17, position 64738741 G → C that targets the $PRKCA$ gene and introduced the D463H amino acid substitution (Fig. 1a, b). The same $PRKCA^{D463H}$ mutation was found in all six ChG tumors from the G2 group (thus 10 of 10). Sanger sequencing analysis of genomic DNA and/or cDNA from the 10 tumors confirmed the presence of the mutant allele in all tumors. The same $PRKCA^{D463H}$ mutation was also found in all the ChGs from the FFPE G3 tumors, except one inconclusive sample, fixed in picric acid—which produced poor-quality DNA and RNA, and may be a false negative (Table 1). The $PRKCA$ gene encodes the PKC alpha protein, which is a member of the AGC (PKA, PKG, PKC) family of cytoplasmic serine/threonine kinases.

Next, we asked whether the $PRKCA^{D463H}$ variant is a truncal alteration in ChG. Clonal analysis was performed by ABSOLUTE algorithm on three G1 samples with sufficient tumor material for copy number variation (CNV) profiling. We estimated an average tumor purity of $0.27 \pm 0.09$, primarily affected by abundant infiltration of immune cells (Supplementary Data 1). Under these conditions, the $PRKCA^{D463H}$ cancer cell fraction was equal to 1 for all cases thus suggesting that the mutation is clonal.

Given the high frequency of the $PRKCA^{D463H}$ mutation in patients with ChG, we next investigated whether the same variant was also present in other tumors occurring in the brain or elsewhere. First, we queried the Catalog of Somatic Mutations in Cancer—COSMIC[6], and cBioPortal database characterizing 31,920 cancer samples and found no $PRKCA^{D463H}$ mutation referenced to date. Furthermore, we sequenced 283 primary brain tumor samples from the Onconeurotek tumor bank that includes several subtypes of gliomas including some that display juxta-

ventricular location or expansion from circumventricular organs and that might be considered to be closely related to ChG. Again, none of the samples we investigated carried the mutation (Supplementary Data 2).

**Genomic landscape of ChG.** We examined the 346 identified somatic mutations in the four G1 samples in order to examine for other recurrent or driver mutations. In contrast to $PRKCA^{D463H}$ that was present in all four G1 samples, the two other recurrent mutations were only present in 2/4 G1 samples and are likely to be passenger and functionally irrelevant changes: (i) $PABPC3$: $L530P$ which is not predicted to have a functional impact (Sift and Polyphen scores) and (ii) chr6:26745595 which is not located within known genes. With the exception of the $SMARCA4$ that was found to be mutated twice at two different intronic positions (both predicted to have no functional impact), no other cancer-related gene was found to be recurrently mutated in the G1 group (Fig. 1b). Ten additional somatic mutations occurring in only one sample were found among Cosmic gene census[6] genes, but only three were missense mutations with putative deleterious functional impact as predicted by Sift[7] or Polyphen[8] score ($FGFR4^{L661V}$, $NCOR1^{P829Q}$, and $CDK12^{S293R}$) and none of them was reported in the Cosmic data base (Supplementary Data 3).

For the G2 group samples, in the absence of germline information, we inspected only sites that were already identified as somatic in the four G1 samples. Beside $PRKCA^{D463H}$ only three of the 346 somatic mutations of G1 group occurred in more than one sample. They were known dbSNP polymorphisms predicted to be non-deleterious.

CNV analysis for G1+2 samples showed recurrent deletions of 16, 17, 19, 22q chromosomes, gain of 4q, and a focal amplification in 18q12.1 which contains $NOL4$ and $ASXL3$ genes (Supplementary Fig. 1; Supplementary Data 4, 5).

The ten samples of groups G1 and G2 were analyzed for expression levels and for pathways using ssGSEA. Among the top cancer-related pathways most significantly enriched in ChG were: upregulation of (i) Pi3K-AKT-mTOR, (ii) EIF2 pathway, and (iii) RAS signaling pathway (the complete pathways measures are given in Supplementary Data 6).

Next, Ingenuity Pathway Analysis (IPA) was performed on differential expression analysis of G1+G2 ChG samples, compared to TCGA grade II wild-type $IDH$ which are closely related tumor with high PKCα expression (Supplementary Fig. 2). The top 10 activated pathways involve the biological processes of translation, proliferation, apoptosis, cell survival, and immune response (Supplementary Data 7 and Supplementary Fig. 3). The most significantly activated pathway was EIF2 pathway ($p$ = 1e−11, Fisher exact test, Supplementary Data 8). EIF2 pathway is critical for translation initiation[9], and of particular interest as PKCα regulates translation by phosphorylating elements of the EIF complex[10–12]. These kinase relations were also confirmed using harmonizome[13] search for PKCα kinase substrates. We therefore performed an IPA analysis of expression data from $PRKCA$ knockout vs. wild-type in umbilical human cells experimental data set (GEO:GSE27869) and found that the EIF2 pathway was activated ($p$ = 0.004, Fisher exact test) in $PRKCA$ knockout cells suggesting that PKCα$^{D463H}$ mutation acts on EIF2 complex as a loss of function. In addition EIF2 pathway regulates activation of RAS and AKT/mTOR pathways (the latter was also among the 10 most significant pathways upregulated in ChG compared to grade II gliomas), thus strengthening the results obtained by ssGSEA described above.

Fusion analysis of the RNA-Seq data revealed no fusion products that were supported by each of the three algorithms (see Methods). Two putative fusion products were supported by two

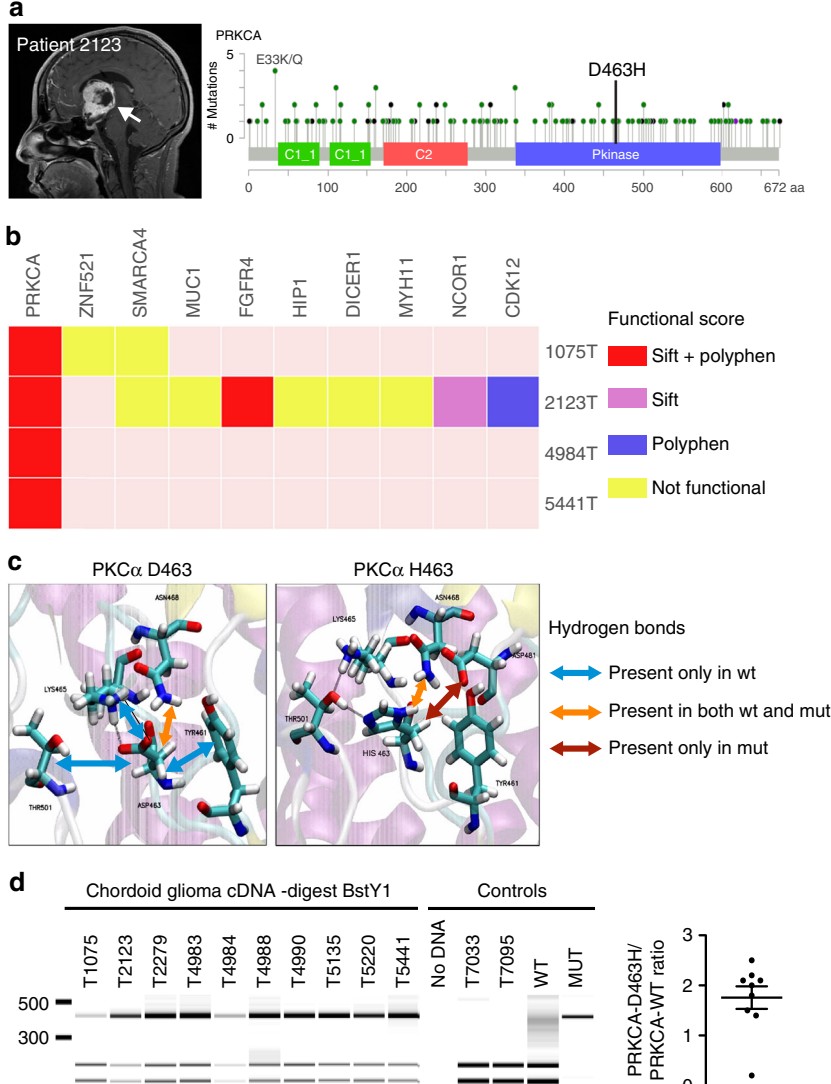

**Fig. 1** Characterization of *PRKCA^D463H^* mutation in chordoid glioma. **a** Left panel: gadolinium T1 weighted MRI showing a well-limited, strongly enhancing mass developed at the anterior part of the third ventricule (patient 2123). Right panel: Pan-cancer *PRKCA* somatic mutations in TCGA cohort. *D463H* was not identified in this cohort and its position is marked by the straight line. **b** Somatic mutations in *PRKCA* in Cosmic census genes for the ChG samples with matched blood (G1 group). The predicted impact of the mutations according to Sift[7] and Polyphen[8] scores are given. **c** Three-dimensional structure of the catalytic site of PKCα showing the positions of D463 in the wild-type, H463 in the mutant protein, and the amino acids predicted to interact via hydrogen bonds with wild-type and mutant 463 residue. Hydrogen bonds identified using CHARMM software and Baker and Hubbard algorithm in MDTraj[70] python package (see Methods) are indicated by arrows: orange (common to both WT and mutant), blue (present only in WT), red (present only in the mutant). **d** *PRKCA^D463H^* transcript is overexpressed in ChG compared to *PRKCA^WT^*. Left: the G to C substitution leads to the loss of BstYI restriction site in the mutant allele. PCR products from ChG cDNA were digested by BstYI then analyzed by LabChIP GX. Lanes T7033 and T7095 correspond to cDNA from low-grade glioma wild type for *PRKCA*, which shows that digestion efficiency is 100%. Mutant product size appears arround 418 bp; wild type digested product sizes appears around 227 and 185 bp. The last two lanes correspond to pcDNA plasmid controls, either WT or D463H mutant (MUT). Right: concentration ratio of mutant to wild-type bands is indicated (mean = 1.8)

algorithms (CLU_PSAP and CLU_VIM) (Supplementary Fig. 4). No fusion involving a COSMIC gene was identified by more than one algorithm.

**Structural analysis of PKCα^D463H^ predicts altered function**. The D463H mutation affects a critical residue of the kinase domain of PKCα (Fig. 1a, c, Supplementary Fig. 5). Aspartate 463 establishes a hydrogen bond and acts as a proton acceptor. CHARMM simulation predicts that hydrogen bonds which are critical for molecule recognition, differ markedly between WT

(TYR461, LYS465, ASN468, and THR501) and D463H mutant (ASN468 and ASP481) (Fig. 1c), and suggests that such conformational changes may modify enzyme/substrate affinity and specificity[14].

Sequence homology analysis identified 392 proteins with kinase domain homologous to PKCα. Of these, 380 have Aspartate in an analog position to the 463 position in PKCα and 6 proteins have Histidine in the 463 analog position. Histidine in this position at the active site of WT proteins may confer distinct properties, for instance substrate preference changes (Supplementary Data 9).

**Table 1 Clinical and genetic feature of the 16 patients with ChG**

| ID | Group | Sex | Age at surgery | WES | RNA-Seq | WES | Sanger PRKCA$^{D463H}$ | | Sanger PRKCA$^{D294}$ |
|---|---|---|---|---|---|---|---|---|---|
| | | | | | | PRKCA 463 | DNA | cDNA | DNA |
| 1075 | 1 | M | 42 | 1 | 1 | mut-CAT | 1 | 1 | 0 |
| 2123 | 1 | F | 55 | 1 | 1 | mut-CAT | 0 | 1 | 0 |
| 4984 | 1 | M | 39 | 1 | 1 | mut-CAT | 1 | 1 | 0 |
| 5441 | 1 | M | 33 | 1 | 1 | mut-CAT | 1 | 1 | 0 |
| 2279 | 2 | F | 45 | 1 | 1 | mut-CAT | 1 | 1 | 0 |
| 4983 | 2 | M | 42 | 1 | 1 | mut-CAT | 1 | 1 | 0 |
| 4988 | 2 | M | 46 | 1 | 1 | mut-CAT | 1 | 1 | 0 |
| 4990 | 2 | M | 65 | 1 | 1 | mut-CAT | 1 | 1 | 0 |
| 5135 | 2 | M | 60 | 1 | 1 | mut-CAT | 1 | 1 | 0 |
| 5220 | 2 | F | 68 | 1 | 1 | mut-CAT | 1 | 1 | 0 |
| 4985 | 3 | M | 63 | 0 | 0 | ND | 1 | NA | 0 |
| 4986 | 3 | F | 56 | 0 | 0 | ND | 1 | 1 | 0 |
| 4989 | 3 | F | 71 | 0 | 0 | ND | 0 | NA | 0 |
| 4991 | 3 | F | 34 | 0 | 0 | ND | 1 | NA | 0 |
| 4782 | 3 | F | 27 | 0 | 0 | ND | 1 | 1 | 0 |
| 9064 | 3 | F | 44 | 0 | 0 | ND | 1 | NA | 0 |

**The PRKCA$^{D463H}$ mRNA is preferentially expressed in ChG.** As all ChGs harboring the PRKCA$^{D463H}$ mutation also retain a copy of the PRKCA$^{WT}$ allele, we sought to determine the relative abundance levels of PRKCA$^{WT}$ and PRKCA$^{D463H}$ products in ChG. To distinguish between PRKCA$^{D463H}$ and PRKCA$^{WT}$ mRNA levels in ChG, we took advantage of the loss of a BstYI restriction site in the mutant sequence. Undigested vs. digested PCR products ratio reflects the abundance of the mutant PKCα mRNA, compared to the WT one. Despite the high level of normal cell infiltration, we observed a mean 1.8× increase of undigested amplicon specifically in the ChG tumor cDNA samples. In the same conditions, low-grade glioma control samples amplicons were completely digested. This indicates that mutant mRNA levels are more abundant in the majority of ChG samples compared to WT mRNA (Fig. 1d, Supplementary Fig. 6, Supplementary Data 10).

**PKCα$^{D463H}$ protein stability and cellular localization.** To determine the functional consequences of the PRKCA$^{D463H}$ mutation, we expressed ectopic myc-tagged PKCα proteins (WT, D463H, and the hypo-active T638A[15]) in human astrocytes. Interestingly, we found that the mutant PKCα$^{D463H}$ protein was less stable than PKCα$^{WT}$, with a half-life of 4.5 vs. 17 h, respectively (Fig. 2a). We next sought to establish whether the mutant protein manifests an aberrant pattern of subcellular compartmentalization. The active wild-type PKCα is typically localized at the cell membrane where it is often enriched within cell–cell junction for the regulation of cell–cell contacts, cell migration, proliferation signals, and contact inhibition[16,17]. As expected, a significant fraction of the WT and hypo-active T638A was detected at the leading edge of the plasma membrane. Conversely, the D463H mutant failed to localize at the plasma membrane and showed diffuse cytoplasmic localization (Fig. 2b and Supplementary Fig. 7). Subcellular fractionation confirmed loss of the PKCα$^{D463H}$ protein from the plasma membrane (Fig. 2c).

We also determined the pattern of subcellular compartmentalization in vivo by comparing the subcellular distribution of PKCα in ChG and normal ependyma. Ependymocytes, which are presumed to be the cell of origin of ChG in the normal brain[3,4], line the human ventricles and are highly polarized cell types with well-organized adherent junctions[18]. In these cells, PKCα accumulated at the sites of cell–cell junctions and colocalized with N-Cadherin, underscoring its regulatory function of cell adhesion in the cells of origin of ChG (Fig. 2d). The staining for PKCα in ChG harboring the D463H mutation showed a different localization pattern, characterized by lack of membrane and cell junction localization, lack of colocalization with N-Cadherin and diffuse cytoplasmic localization (Fig. 2d), thus recapitulating the in vitro findings (Fig. 2b, c). Taken together, the above results indicate that the D463H mutation prevents localization of the PKCα protein at the plasma membrane.

**PKCα$^{D463H}$ enhances cell proliferation.** Expression of ectopic PKCα$^{D463H}$ in human astrocytes resulted in significant stimulation of proliferation, when compared to cells expressing PKCα$^{WT}$, PKCα$^{T638A}$, or a control GFP (Fig. 3a). D463H mutation is specific to ChGs which derive from tanycytes, a specific type of slowly proliferating ependymal cells from the circumventricular organs[4,19]. We therefore investigated the effect of PKCα$^{D463H}$ on tanycytes. We measured EdU incorporation in primary rat tanycytes and observed increased incorporation in cells expressing PKCα$^{D463H}$ compared to PKCα$^{WT}$ ($p < 0.001$, Fisher exact test), thus indicating that the PKCα$^{D463H}$ mutation enhances cell cycle progression (Fig. 3b).

Taken together, our data indicate intriguing properties for the PKCα$^{D463H}$ variant specific to ChG. Those include enhanced cell proliferation, enhanced expression, and reduced protein stability. While hotspot mutation suggests a gain of function, EIF2 pathway activation appears mediated by a loss of normal PKCα function. Hence it is likely that PKCα$^{D463H}$ represents a neomorphic enzyme combining loss of function to novel enzymatic properties.

**Discussion**

We identified PRKCA$^{D463H}$ as a novel recurrent mutation affecting specifically virtually all ChG (15/16 = 94% in our series, with one possibly false negative) and absent in any other brain or systemic tumors analyzed so far.

Since tumors are usually influenced by more than one genetic driver[20], we looked for additional genetic alterations. However, none of them was present in more than two samples (Fig. 1b), suggesting that PRKCA$^{D463H}$ represents a major driver in the tumorigenesis of ChG. Taking into account the significant degree of normal cell contamination of ChG, PRKCA$^{D463H}$ mutation emerges as the clonal and truncal genetic alteration characterizing this disease. The PKCα$^{D463H}$ mutation converts

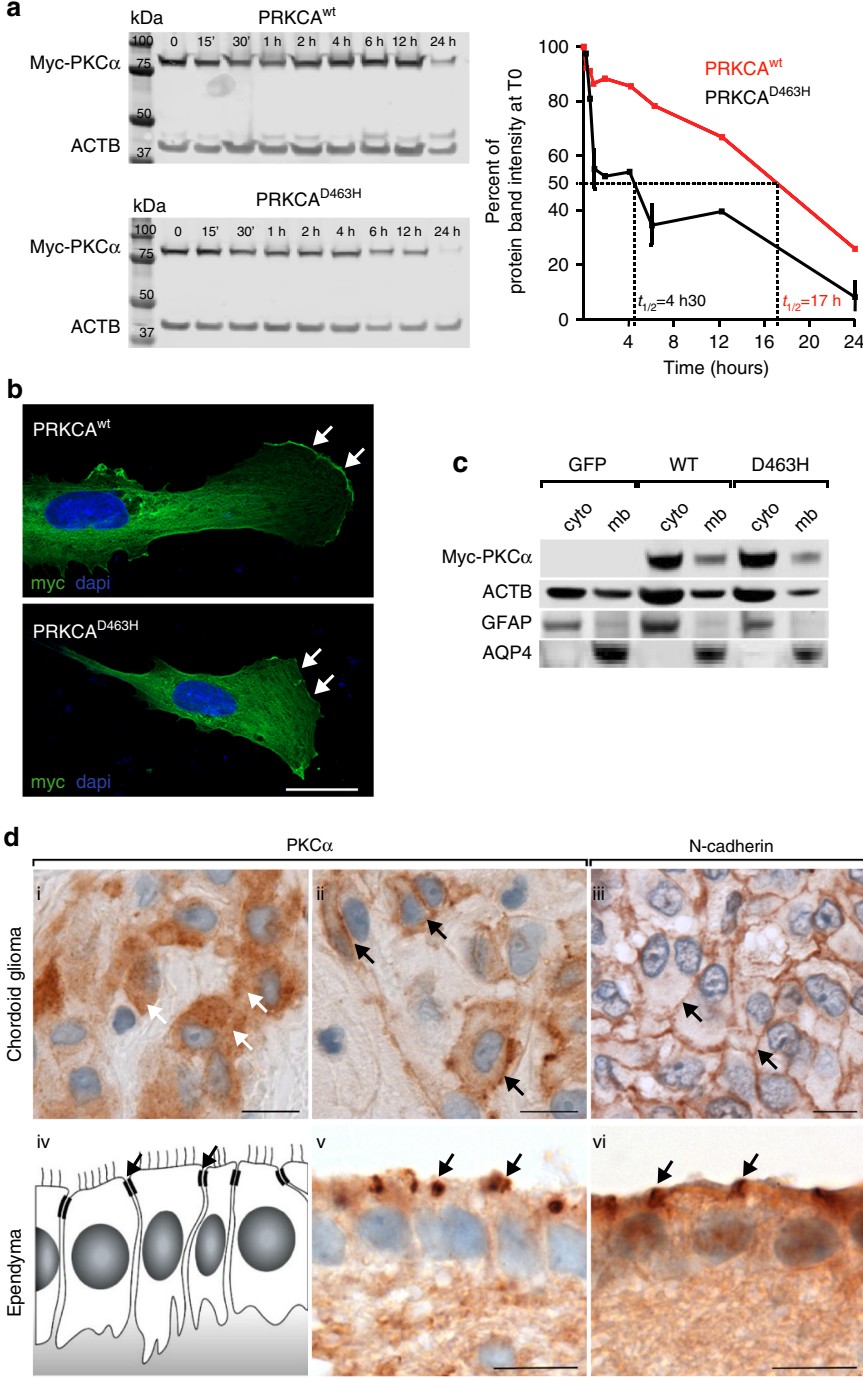

**Fig. 2** Characterization of the PKCα$^{D463H}$ mutant protein. **a** PKCα$^{D463H}$ is less stable than PKCα$^{WT}$: blockage of protein synthesis with 500 µg/mL of cycloheximide was performed on transduced COS-7 cells (see Methods) with dosage of the recombinant protein at different times, and shows decreased half-life of D463H mutant vs. wild-type protein. **b** PKCα$^{D463H}$ mutation leads to membrane delocalization. Human immortalized astrocytes transduced with *PRKCA*$^{D463H}$-Myc lentivirus leads to decreased Myc expression at the membrane, compared to astrocytes transduced with *PRKCA*$^{WT}$-Myc. Scale bar = 10 microns. **c** Subcellular fractionation of transduced cells shows moderately decreased amount of PKCα$^{D463H}$-Myc protein at the membrane, compared to PKCα$^{WT}$-Myc and control GFP lentivirus (ACTB = beta-actin, AQP4 = aquaporin 4). **d** Expression and subcellular localization of PKCα in chordoid gliomas. Subcellular localization of PKCα (brown; panels i, ii, v) is cytoplasmic (i, white arrows), and much less frequently diffusely lining the cytoplasmic membrane (ii, black arrows), similar to N-Cadherin (iii, black arrows). In comparison, in normal ependyma, PKCα is concentrated at the zonula adherens (v, black arrow), similar to N-Cadherin (vi, black arrows). Scale bar = 10 microns

a highly conserved Aspartic Acid in the kinase active site into an Histidine. The D463H amino acid change is predicted by three independent algorithms to harbor major functional consequences that are likely to be related to the key, deprotonating activity of D463 toward the Ser/Thr substrates of PKCα (Protein data base[21]).

*PRKCA* is mutated in a wide range of human cancers, but *PRKCA*$^{D463H}$ has not been reported in other tumors, including

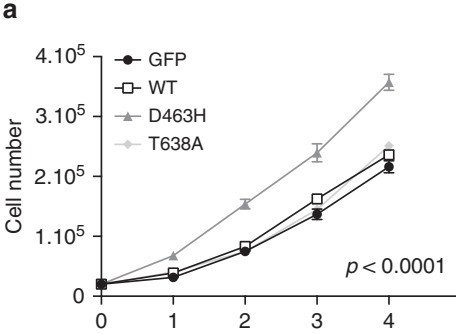
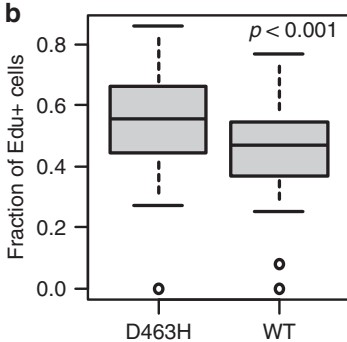

**Fig. 3** Effect on cell proliferation. **a** *PRKCA^D463H^* mutation enhances proliferation of human astrocytes. Human immortalized astrocytes overexpressing PKCα^D463H^-Myc grow significantly faster ($p < 0.0001$; ANOVA) than astrocytes overexpressing PKCα^WT^-Myc, PKCα^T638A^-Myc, or GFP. This analysis was performed from duplicate plates. **b** Primary rat tanycytes transduced with *PRKCA^D463H^* have a higher DNA replication rate than those transduced with *PRKCA^WT^*. Myc-positive cells, and Myc-positive–EdU-positive cells were counted. Results are from three independent experiments. The boxplots indicate the median with the box bounded by the 25 and the 75 percentiles

those more closely related to ChG (Supplementary Data 3), thus suggesting that the D463H mutation is likely to harbor a specific impact in the cell and tissue microenvironment that sustains ChG tumor initiation. Interestingly, two other brain tumors harbor a specific signature affecting the *PRKCA* gene: the *SLC44A1-PRKCA* gene fusion in papillary glioneuronal tumors and the *PRKCA^D294G^* in pituicytomas, another tumor from the diencephalic region of the brain[22]. ChG and pituicytomas share several similarities such as TTF1 expression and mTor pathway activation[23]. However, we found no overlap between the two entities since (i) *PRKCA^D463H^* was absent in the pituicytomas we analyzed (Supplementary Data 3) and *PRKCA^D294G^* was absent in ChG (Table 1), and (ii) *PRKCA^D294G^* does not affect the active site but rather impairs the membrane attachment at the inter-cellular junctions[22,24]. In addition, both mutants are characterized by a defect in membrane anchorage.

Since D463H mutation is highly specific to ChG, we chose to test its effects on cellular models as close as possible to this tumor physiological context. Our results show that PKCα^D463H^ ectopic expression results in increased proliferation rate of immortalized human astrocytes and increased DNA replication in primary rat tanycytes.

PKCα is involved in a wide range of cellular processes (mitosis, cell adhesion, polarity, and migration), and, depending on the cellular context, can either act as an oncogene or as a tumor suppressor gene[25–28]. This duality is also supported by our data in the particular context of ChG. To the best of our knowledge, there is no example in oncology of a highly recurrent missense mutation affecting the active site of an enzyme and resulting merely in the loss of enzymatic activity. Several data further suggest that the D463H hotspot mutation results in a gain of function. Comparison of the abundance of *PRKCA^WT^* and *PRKCA^D463H^* transcripts within the same tumors showed mutant mRNA levels prevalence. Noteworthy, PKCα has been shown to be involved in contact inhibition[17], and PKCα inhibition promotes the formation of cell junctions[29]. Finally, while the Aspartate to Asparagine substitution leads to a PKCβII^D466N^ catalytically inactive[30], Histidine stay at a position homologous to the Aspartate residue as a proton acceptor in a number of native enzymatic active sites (see Supplementary Data 9), further supporting the gain of function hypothesis.

On the other hand, our computational analysis show a consistent and robust activation of EIF2 pathway (and to a lesser extent AKT/mTor and Ras pathway) in *PRKCA^D463H^*-mutated ChG. EIF2 activates the initiation of translation and is itself downregulated by several kinases, including PKCα[10,11,31,32].

Intriguingly, both upregulation and downregulation of EIF are involved in oncogenesis[32]. The fact that PKCα inactivation is associated with upregulation of EIF2 strongly suggests that upregulation of EIF2 in ChG is related to the loss of the normal function of PKCα.

Our data illustrate the multifaceted functions of PKCα. Neomorphic mutations were reported for several cancer-related genes[33] and specifically also for *PRKCG*[34]. The *PRKCA^D463H^* may lead to a new neomorphic PKC that modify substrate preferences while altering its normal function. Moreover, those functions seem to depend also on the cellular context, as supported by the fact that *PRKCA^D463H^* which is almost constant in ChG, have never been reported in any other tumor.

PKCα represent an actionable target. PKCα inhibitors have been unsuccessful in most cancers including glioblastomas, probably because loss of function was the prevalent mechanism in these cases[26]. In the case of ChGs, the inhibition of the neomorphic PKCα^D463H^ could open a new perspective for specific targeting of this rare subtype of glioma. Interestingly, during the revision process of this paper, the same *PRKCA^D463H^* mutation, found in 13/13 ChGs, was reported and was associated with activation of Ph°-ERK and sensitivity to Ph°-ERK inhibitor[35].

## Methods

**Patients**. ChG were recruited from the Pitié-Salpêtrière tumor bank Onconeurotek and the RENOP network (Réseau de Neuro-Oncologie Pathologique)[3]. Diagnosis was confirmed by histological review by two pathologists (F.B. and K.M.). The patients' informed consent and ethical board approval were required for collection of tumor samples and clinical-pathological information, as stated by the Declaration of Helsinki.

**Nucleic acids extraction**. DNA from cryopreserved (10/16) or FFPE tumor samples (6/16) was extracted using the QIAamp DNa Mini Kit (51304, Qiagen) or iPrep™ ChargeSwitch® Forensic and Buccal Cell Kits (IS-10002 Life Technologies). RNA was extracted from frozen tissue (10/16) using iPrep™ Trizol® Plus RNA Kit (IS-10007 Life Technologies) and from FFPE samples (2/16) using RNeasy FFPE Kit (73504, Qiagen) according to the manufacturer's instructions. DNA were extracted from blood samples using NucleoSpin® Blood L (740954, Macherey-Nagel).

**Sanger sequencing**. Separate primers were designed for FFPE and frozen samples (seq primer in Supplementary Data 11). PCR were performed in the following conditions: 94 ℃ for 3 min, 45 cycles of 94 ℃ × 15 s, 60 ℃ × 45 s and 72 ℃ × 1 min, and a final step at 72 ℃ for 8 min.

The PCR products were purified using NucleoFast® 96 PCR plates and NucleoVac 96 Vacuum Manifold (Macherey-Nagel) or Agencourt® AMPure® XP (Beckman Coulter) for short amplicons, and submitted to Sanger sequencing. The forward and reverse sequences were visualized using Chromas Lite software.

**Copy number variation analysis**. CNV analysis was performed using the Illumina HumanCoreExome-24 array according to manufacturer protocols. Normalized intensity signals were generated from the Illumina GenomeStudio software. We used CBS[36] for CNV analysis. GISTIC2.0[37] was utilized in order to identify significantly repeated CNV.

**Exome sequencing**. Exome sequencing was performed for DNA from frozen material ($N = 10$) and paired blood samples when available ($N = 4/10$). Library was prepared using Kapa Library Preparation Kit for Illumina Sequencing System (07137974001, Roche) and exome capture using SeqCap EZ Exome Enrichment Kit v2.0 (44 Mb 05860504001, Roche) paired-end (2*75 bp) for 13 samples and by SeqCap EZ Exome Enrichment Kit v3.0 (2*150 bp) for one blood sample. Paired-end sequencing was performed by HighOutput kit on NextSeq500 (Illumina).

The data analysis consisted of the following steps: (1) adapter sequence was removed using Cutadapt[38], (2) poor-quality reads were removed using Trimomatics[39], (3) alignment was performed by BWA[40] with hg19 as the reference genome, (4) quality control was performed by Qualimap[41], (5) deduplication was done using samtools[42], (6) variant calling was performed using GATK pipeline[43,44] for the samples without germline information ($N = 6$), (7) somatic mutation analysis was done using Mutect[45] for the four samples with germ line information, we excluded potential Covaris-induced mutations as per Costello et al.[46] using in-house scripts[47], (8) annotation was performed using Oncotator[48] including mutation functional prediction by Sift[7] and Polyphen[8].

**RNA sequencing**. Libraries were generated from total RNA using TruSeq Stranded mRNA LT Kit (Illumina RS-122-2101). Paired-end sequencing ($2 \times 150$ bp) was performed by Nextseq500 using High Output kit. The data analysis consisted of the following steps: first, the quality of raw reads has been assessed with FastQC[49]. Trimming was done with Trimmomatic[39]: adapters were cut and reads <40 bp were excluded. The processed reads were aligned on human reference genome hg19 with Tophat2[50]. The expression levels were evaluated by CuffLinks. Fusion identification and analysis were performed by three different algorithms: (1) Tophat-fusion[51], (2) FusionCatcher[52], (3) PRADA[53]. We excluded the putative fusion pairs which did not fulfill the following minimal criteria: (i) at least two read pairs aligned to both sides of the breakpoint and (ii) at least one read aligned on the breakpoint.

**Clonal analysis**. Clonal analysis was performed using Absolute[54], using CNV profiles and somatic mutation profiles that were produces by CBS[36] and MuTect[45], respectively. We used the program default parameters and also reviewed manually the selected estimated solutions.

**Pathway analysis**. Pathway analysis was performed using single sample Gene Set Enrichment Analysis (ssGSEA)[55] using GenePatterns[56]. We used Biocarta, PID, and KEGG pathways lists from MsigDB[57]. Pathway analysis was also carried out for the differential expression analysis of the 10 G1+G2 ChG samples and the grade II glioma IDH wt cohort of TCGA. The raw counts from both cohorts were jointly normalized using the EDASeq protocol with the full-quantile option in the between normalization step[58]. Differential expression analysis was carried out using R's LIMMA package[59]. The pathway analysis was performed for the top 2000 most significant differentially expressed genes, using IPA[60]. The top 600 most significant differentially expressed genes in the PRKCA knockout vs. wild-type in umbilical human cells (GEO: GSE27869) were obtained using Harmonizome[13].

**Structural analysis of mutant PKCα**. PKCα's structure and homology modeling of the mutant: The crystal structure of PKCα[wt] was available in RCSB database (www.rcsb.org21: PDB ID 3IW4). The structure of the mutant was produced using homology modeling server named I-TASSER (Iterative Threading ASSEmbly Refinement)[61–64].

CHARMM for molecular simulations: CHARMM[65] was used to produce 70 nanoseconds of molecular simulation in water for PKCα[wt] and PKCα[D463H]. The analysis was performed at pH 7 (characteristic of Gliomas[66]). CHARMM scripts were obtained using CHARMM-GUI (http://www.charmm-gui.org). CHARMM calculates and predicts protein's different conformations around its stable conformation using leapfrog verlet integrator algorithm[65]. While looping through each conformational frame in PKCα[wt] and PKCα[D463H] simulations, MDtraj was used to identify hydrogen bonds involving D463 and H463[67]. Baker and Hubbard algorithm identifies hydrogen bonds using their respective atom distances and atom angles for each frame[67]. The threshold of atom angles/distances must be respected in at least 80% of total number of the frames by default.

Protein sequence homology: Position-Specific Iterated BLAST (PSI-Blastp)[68] was used to find homologous proteins for PKCα. The query sequence was the kinase domain sequence of the mutant PKCα. Alignment parsing and analyzing scripts written in python were used to identify the position and amino acid homologous to position 463 of PKCα in the homologous proteins.

**RT-PCR**. RT-PCR detection of $PRKCA^{D463H}$ mutant cDNA: cDNA were retro-transcribed from tumor RNA samples (1 µg) with Maxima First Strand cDNA Synthesis Kit for RT-qPCR (Thermo Fisher Scientific K1641) according to manufacturer instructions, then amplified with PCR primers VG-229 and VG-623 by using Fast Start PCR Master (Roche 04710452001). PCR reactions were split and half was digested with BstYI (New England Biolabs), when the other half was incubated in the digestion mix without BstYI, and left over night at 60 °C. Fragments were analyzed on 2% agarose gel or by using LabChip GX bioanalyzer (Caliper LifeSciences, Villepinte, France).

$PRKCA$ RT-qPCR was performed using LightCycler® 480 Probes Master mix and Universal Probe Library™ probes specific to each genes. Primers and probe number are indicated in Supplementary Data 11. The reference primers amplified PPIA (Peptidylprolyl isomerase A = Cyclophilin A). Real-time qPCR reactions were performed according to the manufacturer's instructions. The $2^{-\Delta CT}$ method was used to determine the relative expression, where $\Delta CT = CT_{target\ gene} - CT_{PPIA}$.

$PRKCa^{WT}$ cDNA synthesis for plasmid construction (see below): RNA was extracted from a glioblastoma spheroid culture, non-mutated on $PRKCA$. First Strand cDNA Synthesis Kit (Fisher Scientific, Illkirch, France) was then used to generate cDNA.

**Immunohistochemistry**. FFPE sections (4 microns) of lateral ventricle ependyma and ChGs were obtained from Pitie-Salpetriere hospital and other centers of the RENOP (Réseau de Neuro-Oncologie Pathologique) network. Deparaffinization and immunolabeling of the sections were performed with a fully automatic system, Ventana benchmark XT System (Roche, Basel, Switzerland) using streptavidin–peroxidase complex with diaminobenzidine as the chromogen. Mouse monoclonal anti-PKCα (ref. Sc-8393 Santa Cruz Biotechnology) diluted at 1/800 was incubated at 37 °C during 32 min, rabbit polyclonal anti-N-Cadherin ((H-63) sc-7939, Santa Cruz Biotechnology) diluted at 1/50 was incubated at 37 °C during 32 min.

**Lentivirus constructs**. $PRKCa^{wt}$ cDNA was amplified from human total cDNA preparation (see above) by PCR by using KAPA HiFi HotStart ReadyMix PCR Kit (KK2601, Cliniosciences) with adapter primers XhoI-PRKCA-F and SacII-PRKCA-R, then cloned after XhoI/SacII restriction and ligation into pcDNATM3.1/myc-His B plasmid (V800-20, Thermo Fisher Scientific), to construct the $PRKCa^{wt}$ pcDNA plasmid. The D463H missense mutation was substituted with PCR mutagenesis with primers MD pcDNA-D463H-F and MD pcDNA-D463H-R on the wild-type $PRKCA$ pcDNA plasmid by using Q5 Site-directed Mutagenesis kit (New England Biolabs E0554S) (seq primer in Supplementary Data 11).

PCR products containing the CMV promoter and $PRKCa^{wt}$ and $PRKCa^{D463H}$ cDNA sequences fused to myc/His tag were amplified from pcDNA expression vectors with adapter primers attB1-CMV-F et attB2-pcDNA3-R, then pEntry vectors were generated with Gateway BP Clonase II Enzyme Mix (11789, Thermo Fisher Scientific) by Gateway recombination (Life Technologies 12535). The T638A missense mutation was substituted by PCR mutagenesis with primers MD pcDNA-T638A-F and MD pcDNA-T638A-R on the wild-type $PRKCA$ pEntry vector by using Q5 Site-directed Mutagenesis kit (New England Biolabs E0554S). Lentiviral constructs were then cloned from the subsequent pEntry vectors by Gateway recombination into a pDEST 75 vector (pDEST-RfAsens-IRES-Puromycin-dU3, kind gift from P. Ravassard). The final pTRIP lentiviral constructs bear a *neomycin* gene and confer resistance to G418. All plasmids were sequenced before use.

**Cell culture**. Immortalized human adult (Clonetics, Walkersville, MD) and fetal (ABM, T0281) astrocytes were used and cultured in DMEM 10% FBS and Prigrow IV medium (TM004, ABM), respectively.

To assess the effect of mutant PKCα on cell proliferation, 150,000 astrocytes were infected with control (GFP), $PRKCA^{WT}$, $PRKCA^{T638A}$, and $PRKCA^{D483H}$ lentivirus for 6 h in the presence of 4 µg/ml of polybrene. Cells were trypsinized 48 h later and plated at 20,000 cells/well in triplicate in 6-well plates (for the growth curve). 12 h later cells were washed in serum-free DMEM twice and cultured in 0.2% for the duration of the growth curve or for 48 h for the western blot (see below). Cell number was assessed by trypan blue exclusion. Three independent infections were performed.

**Immunofluorescence on cells**. For immunostaining, cells were fixed in PBS, 4% paraformaldhehyde (15710 Electron Microscopy Sciences) 5 min at room temperature (RT), washed in PBS and permeabilized with PBS, 0.25% Triton X-100 5 min at RT. After three washes in PBS, cells were blocked with PBS, 20% goat serum 1 h at RT. Primary antibody (anti-Myc; Cell Signaling, #2276 diluted 1/500) was added to cells in PBS, 2% goat serum for 2 h at RT under gentle rocking. Cells were washed three times with PBS then incubated with the secondary antibody (Alexa-488-conjugated anti-mouse IgG1; Molecular probes; diluted 1/500) together with DAPI (0.15 ng/ml) 1 h at RT. Cells were then washed three times with PBS and mounted in Fluoromount (Sigma F4680). Pictures were acquired on an Apotome1 microscope (Zeiss).

**Subcellular fractionation and western blot**. Astrocytes were transduced with control (GFP), $PRKCA^{WT}$, $PRKCA^{T638A}$, and $PRKCA^{D483H}$ lentivirus in the

presence of polybrene. After 48 h, cells were plated in 60-mm collagen-coated dishes (400,000 cells per dish) in medium containing 0.2% FBS. Cells were harvested 24 h later. Total cell extracts were prepared by lysing cells with RIPA (Pierce, Brebieres, France) supplemented with protease and phosphatase inhibitor cocktail (Thermo Fisher #78440). Membrane and cytosolic subcellular fractions were isolated using a subcellular fractionation kit (Thermo Fisher #78840).

Western blot was performed according to standard protocols. The following primary antibodies were used: anti-PKCα (Santa Cruz, SC-8393; diluted 1:500), anti-Myc (Cell Signaling, #2276; diluted 1:1000), anti-β-actin (Cell Signaling, #3700; diluted 1:1000), anti-Aquaporin 4 (Sigma, #HPA014784; diluted 1:500), anti-GFAP (Sigma, #G3893; diluted 1:1000). Secondary antibodies (Odyssey IRDye 800CV goat anti-rabbit or mouse secondary antibodies, Science Tec, Courtaboeuf, France) were diluted 1:5000 in SuperBlock Blocking Buffer in TBS (Pierce, Brebieres, France). Blots were scanned and quantified on the Odyssey CLx (Science Tec, Courtaboeuf, France). Quantification values were normalized to the corresponding β-actin band. The unscropped scans (raw data western blots) are provided in the Supplementary Information file (Supplementary Fig. 8).

**Tanycytes isolation, culture and EdU counting**. Tanycytes were isolated from 10 days old rat as previously described[69] and cultured in DMEM/F12 (31966, Thermo Fisher Scientific) supplemented with 10% (v/v) donor calf serum (Invitrogen), 1% (v/v) L-glutamine (Thermo Fisher Scientific), and 2% penicillin/streptomycin. The tanycytes were plated on inserts (Grenier bio-one 665641) at a density of 442 cells/mm$^2$ and were transduced the same day with the PKCα$^{WT}$ (MOI 10) and PKCα$^{D463H}$ (MOI 20) lentiviruses in the presence of 1 μg/ml of polybrene. The following day the medium was changed. Two days post transduction EdU (1.6 μg/ml) was added to the medium; 12 h later (day 3 post transduction) the cells were fixed and processed for immunostaining as described for astrocytes. EdU staining was chemically revealed with the Click-iT plus kit (C10640, Thermo Fisher Scientific).

**Cycloheximide chase assay**. COS-7 cells stably expressing GFP, PKCα$^{WT}$, and PKCα$^{D463H}$ were used and cultured in DMEM 1 g/L glucose, 10% FBS. To assess the stability of the mutant form of PKCα during time, COS-7 were plated in 6-well plates and incubated for different time with 500 μg/mL of cycloheximide. Total cell extracts were collected at different time points and prepared by lysing cells with RIPA (Pierce, Brebieres, France) supplemented with freshly added protease and phosphatase inhibitor cocktail (Thermo Fisher #78440). The results were analyzed by the SDS-PAGE assay and western blotting using anti-myc antibodies with 1:2500 dilution; β-actin antibodies with 1:10,000 dilution. The bands of Myc and β-actin were quantified in triplicates using ImageJ software.

**Data availability**. All the genomic data sets are available at the European Genome-phenome Archive (EGA), which is hosted by the European Bioinformatics Institute, under the accession number: EGAS00001002433. The original autoradiographies are available as supplementary file (Supplementary Fig. 8). All other remaining data are available within the article and Supplementary files, or available from the authors upon request.

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

## Acknowledgements

Supported by grants from the Ligue Nationale contre le Cancer (M.S., E.H.), Fonds Carnot (M.S.), Hadassah-France (S.R.), Cancéropole IdF (M.S.), Fondation ARC pour la recherche sur le cancer (PJA 20151203562 to F.B.; PJA 20131200481 and PJA 20151203259 to E.H.), FP7 Marie Curie CIG (to E.H.). The research leading to these results has received funding from the program "investissements d'avenir" ANR-10-IAIHU-06. We are indebted to Gabriel Viennet, Anne-Marie Bergemer-Fouquet, Alain Nere for providing samples; Justine Guegan, Karim Labreche, Ludovic Prevost, Philippe Domineaux, Pierre de la Grange for bioinformatics support and advises; Philippe Ravassard for providing the pDEST-RfAsens-IRES-Puromycin-dU3 construct.

## Author contributions

F.B., M.P., D.F.-B., M.-H.A.-L., C.V., A.V., E.L.-Z., M.K., and K.M. collected the ChG cases. F.B. and K.M. performed the centralized review. M.D., M.G., and Y.M. performed NGS and Sanger sequencing. S.R., A.N., and S.P. performed the computational biology analyses. I.S. made the lentivirus constructs. I.S., F.B., V.G., S.P., I.L.R., and A.L. performed the IF and IHC studies. M.V., B.B., and M.D. performed the cellular fractionation and cycloheximide experiments. A.L. performed the cell growth analysis. I.L.R. and A.S. performed tanycytes analysis. S.R., F.B., and M.S. conceived the project. A.I., A.L., E.H., and M.S. supervised the project and discussed the results. S.R., A.I., E.H., and M.S. wrote the manuscript. All the authors read, edited, completed, and approved the manuscript.

## Additional information

**Competing interests:** The authors declare no competing interests.

