## [Peer Review File · Nature Communications]

Reviewers' comments:

Reviewer #2 (Remarks to the Author):

This interesting manuscript entitled "A novel mutation in PRKCA is the major driver of Chordoid Gliomas" by Shai Rosenberg and colleagues is the first report on a recurrent and highly entity-specific PRKCA hotspot mutations in a series of 15 chordoid gliomas. The authors found this novel hotspot mutation in 14 of 15 cases. The authors further provide basic functional data on the newly identified mutation.

Prior to a potential publication, the following issues should be addressed:

The figures in the uploaded pdf are of very poor quality and should as such be replaced.

It would be interesting and relevant to add estimated tumor purity and allele fractions for the PRKCA mutation to Table 1 for each case.

Could the authors further comment on the potential relevance of the intronic SMARCA4 mutations?

The CNV analysis doesn't appear to be very thoroughly done. It is not clear, which of the datasets were taken into account for this and whether the authors identified any (recurrent) focal CNVs?

Why was EIF2 phosphorylation not tested in the functional work-up, which seems an obvious experiment to do?

Could the authors explain, why the overexpression experiment coupled with a proliferation assay was done in normal astrocytes, since these do not seem to be a particularly likely cell of origin?

In contrast, the authors refer to tanycytes as the potential cells of origin for choroid gliomas. Can this be further substantiated with the molecular data obtained from this series?

The manuscript would benefit from critical proofreading by a native speaker.

Reviewer #3 (Remarks to the Author):

In the manuscript by Rosenberg et al, the authors describe a commonly occurring PRKCA (protein kinase C alpha) mutation (D463H) in chordoid gliomas. They describe the mutant as being more highly expressed than WT PKCa, having increased activity in an in-cell assay measuring phosphorylation of the downstream substrate MARCKS, and having depletion of membrane localization. They link the increased activation of the mutant to increased activation of the EIF2 pathway. This is an important and exciting discovery, however several deficiencies need to be addressed prior to publication as follows:

Major comments:

1. It is essential the authors perform a properly controlled in vitro kinase assay comparing WT PKC alpha, the D463H mutant, and a more traditional kinase dead mutant where the Lys of the VAIK motif is mutated, K368M. Preferably this would be performed with purified WT PKCa, D463H, and K368M with equal levels of each mutant and the WT kinase. In general, mutation of the D in the "HRD" motif (YRD in PKCa) will result in loss of catalytic activity, so the authors need to convincingly demonstrate the D463H mutant to be a GOF mutant.

2. Total PKC levels need to be included for Figure 3A. What is the phosphorylation status of the

D463H mutant at the activation loop, turn motif, and hydrophobic motif? All three sites generally need to be phosphorylated for PKC to be fully active. This also needs to be shown for this figure.

3. A mutation at this exact residue was previously characterized (D463N – Antal CE et al, *Chemistry*, 2014), and defined to be in a constitutively open conformation, have decreased phosphorylation, and increased degradation. It is essential the authors examine by western blot the PKC α levels in patient samples. It is highly plausible that no or very little PKC α is expressed at the protein levels indicating this would be a loss-of-function mutation. The authors have only examined expression at the level of mRNA from patients' tumors. The phosphorylation status and protein levels of the endogenous mutant from the tumor cells from the patients also needs to be examined and included in this manuscript.

4. In addition, if cells harboring the mutant were available depletion of the D463N to show decreased proliferation would be more convincing than an overexpression experiment. Correcting the D463N mutant back to WT (via CRISPR/CAS) to show a loss of tumorigenic phenotypes would be the most convincing and compelling approach.

Minor comments:

1. The results that the mutant D463H is localized to cytosol are not novel and expected given the results from the previous study (Antal CE et al, 2014). The authors results should be conveyed in light of the results from this previous publication.

2. A time course experiment should be conducted to assess if the D463H mutant is less stable than WT (ie this could just be an overexpression experiment examining expression levels at 24, 48, and 72 hours to compare relative levels of expression of WT vs D463H mutant and assess if the D463H is less stable).

3. It is surprising that Myc-tagged D463H is migrating at a slower rate than WT (Figure 3A)? It suggests this mutant has a higher level of phosphorylation, but the WT should also be migrating at a slower rate unless it has an inactivating mutation? Also WT overexpression will generally result in an increase in MARCKS phosphorylation?? It is almost as if these two lanes were mixed up? Based on the previous report, the D463H should also migrate as a doublet.

4. Traditional semi-quantitative RT-PCR and sanger sequencing should be used to verify increased expression of the D463H compared to WT at the mRNA level (Figure 1D). When comparing expression of the mutant vs WT were both bands utilized to demonstrate the mutant is expressed at higher mRNA levels? Combining both bands would seem to indicate equal levels of the WT and mutant?

5. PKC is a member of the AGC family of kinases – line 96.

Reviewer #2 (Remarks to the Author):

This interesting manuscript entitled "A novel mutation in PRKCA is the major driver of Chordoid Gliomas" by Shai Rosenberg and colleagues is the first report on a recurrent and highly entity-specific PRKCA hotspot mutations in a series of 15 chordoid gliomas. The authors found this novel hotspot mutation in 14 of 15 cases. The authors further provide basic functional data on the newly identified mutation.

Prior to a potential publication, the following issues should be addressed:

The figures in the uploaded pdf are of very poor quality and should as such be replaced.

This is due to Nature's pdf creation. High quality figures were not required for initial submission. We replaced with high quality pictures in this revised version.

It would be interesting and relevant to add estimated tumor purity and allele fractions for the PRKCA mutation to Table 1 for each case.

We thank the reviewer for this comment and added the tumor purity estimates (as calculated by a leading algorithm – ABSOLUTE) to the Table S1 in the revised version.

Could the authors further comment on the potential relevance of the intronic SMARCA4 mutations?

The two intronic mutations in SMARCA4 gene are predicted to have no functional impact and are likely to be passenger events not carrying functional consequences. They are reported here as this is the sole occurrence of another gene beside PRKCA that is recurrently mutated in our tumor dataset. This point is now clarified in the text (p 6)

The CNV analysis doesn't appear to be very thoroughly done. It is not clear, which of the datasets were taken into account for this and whether the authors identified any (recurrent) focal CNVs?

The CNV analysis was performed using state-of-the art algorithms (GPHMM segmentation for each sample and GISTIC analysis for identification of recurrent CNV areas which is the standard in the literature). The significance of the findings is however dependent on the number of cases, which is of limited size in the chordoid glioma cohort analyzed here. In any case we did not find any clear recurrent finding.

Why was EIF2 phosphorylation not tested in the functional work-up, which seems an obvious experiment to do?

The activity of EIF2 is negatively regulated by the phosphorylation of the Ser51 residue of EIF2alpha. While PKC activation has been associated with EIF2 α phosphorylation (Pushpanjali, Biochimica et Biophysica Acta 1800 (2010) 518–525), it is important to note that the EIF2 α ser51 is not a direct substrate of PKC α (Dey, PNAS, 2011, 108, 4316–4321). The link between PKC α and EIF2 activation is therefore indirect and probably cell context dependent: clearly the determination of the chains of events linking these two effectors is beyond the scope of this manuscript. EIF2gamma (8 sites of phosphorylation) is a potential target of PKC (Andaya, J. Proteome Res. 2011, 10, 4613–4623) however, no antibodies are available for these phosphorylation sites, and their role in EIF2 activation remains elusive.

Therefore, to address the point raised by the reviewer, we considered EIF2alpha ser51 phosphorylation which is not a PKC α substrate, but may reflect EIF2 activation. In line with the results obtained by computational analysis, we observed a defect of Ser51-EIF2 α phosphorylation in astrocytes transduced with PRKCA^{D463H} mutant or the hypoactive mutant PRKCA^{T638A} compared to PRKCA^{WT} (figure 3a of the revised manuscript).

Could the authors explain, why the overexpression experiment coupled with a proliferation assay was done in normal astrocytes, since these do not seem to be a particularly likely cell of origin?

We agree that such a specific mutation should be investigated in a cellular model that better represents the cell-of-origin of Chordoid glioma. The most credited cells-of-origin of Chordoid glioma are tanycytes, which are poorly characterized slowly proliferating cells lining the third ventricle (REF). Tanycyte cultures are challenging mostly because these cells can hardly proliferate in vitro. Nevertheless, to fully address the request of the referee, we adapted a method for the short-term culture of primary tanycytes isolated from newborn rat brain. We used these cultures to transduce either wt or D463H mutant PRKCA and found that, compared with wt PRKCA, the ectopic expression of the D463H mutant protein significantly increased the fraction of tanycytes that enter the S phase of the cell cycle, as measured by EdU incorporation (three different experiments, Fisher exact test: $p < 0.0007$). These results, which were obtained in the most accurate cell-of-origin model of chordoid glioma, confirm that the D463H mutation of PRKCA enhances proliferation and they are presented in Fig. 3e of the revised manuscript.

In contrast, the authors refer to tanycytes as the potential cells of origin for choroid gliomas. Can this be further substantiated with the molecular data obtained from this series?

We characterized in a previous paper (Bielle, Am J Surg Pathol (2015) 39:948-56) Chordoid gliomas with several immunohistochemical antibodies and showed in particular a strong expression of TTF1 transcription factor which is involved in the development of the ventral forebrain and is strongly expressed by the tanycytes from the circumventricular organ of the lamina terminalis. Previous studies have documented similar ultrastructural features in embryonic tanycytes and chordoid gliomas (Sato, Acta Neuropathol (2003) 106: 176-180).

The manuscript would benefit from critical proofreading by a native speaker.

The revised paper has been edited by a native-speaker.

Reviewer #3 (Remarks to the Author):

In the manuscript by Rosenberg et al, the authors describe a commonly occurring PRKCA (protein kinase C alpha) mutation (D463H) in chordoid gliomas. They describe the mutant as being more highly expressed than WT PKCa, having increased activity in an in-cell assay measuring phosphorylation of the downstream substrate MARCKS, and having depletion of membrane localization. They link the increased activation of the mutant to increased activation of the EIF2 pathway. This is an important and exciting discovery, however several deficiencies need to be addressed prior to publication as follows:

Major comments:

1. It is essential the authors perform a properly controlled in vitro kinase assay comparing WT PKC alpha, the D463H mutant, and a more traditional kinase dead mutant where the Lys of the VAIK motif is mutated, K368M. Preferably this would be performed with purified WT PKCa, D463H, and K368M with equal levels of each mutant and the WT kinase. In general, mutation of the D in the "HRD" motif (YRD in PKCa) will result in loss of catalytic activity, so the authors need to convincingly demonstrate the D463H mutant to be a GOF mutant.

We agree that K368M is a more traditional and inactive kinase dead mutant. However T638A has a very low activity (36-22% of WT-PKCalpha, Swanson, J biol chem 289, 17812-17829). T638 is an essential priming phosphorylation site (turn motif) essential for full catalytic activity after activation.

We performed kinase assay on transduced astrocytes (Figure 3b) and used staurosporin, a PKC α inhibitor currently used in clinical trials. While this experiment fails to show a quantitative increase in total phosphorylation it clearly indicates that there is no loss of function: both WT and mutant PKCa result in an increase of kinase activity compared with the GFP control. It is likely that this mutation result in more subtle-qualitative-changes (possibly creating a neomorphic enzyme that modifies substrate specificity). Interestingly, the mutant, but not the WT, appears resistant to Staurosporin inhibition (which precludes a therapeutical use of this agent in chordoid gliomas).

2. Total PKC levels need to be included for Figure 3A. What is the phosphorylation status of the D463H mutant at the activation loop, turn motif, and hydrophobic motif? All three sites generally need to be phosphorylated for PKC to be fully active. This also needs to be shown for this figure.

We show here that there is no loss of phosphorylation at any phosphorylation sites, but instead an increase in the phosphorylation at the turn motif (see figure 3a the ratio phosphoThr638/Myc), and also at the activation loop (Thr497) and hydrophobic motif (Ser657). This suggests that the PKC α ^{D463H} is mostly in a mature state and prone to activation.

3. A mutation at this exact residue was previously characterized (D463N – Antal CE et al, Chem biology, 2014), and defined to be in a constitutively open confirmation, have decreased phosphorylation, and increased degradation. It is essential the authors examine by western blot the PKCa levels in patient samples. It is highly plausible that no or very little PKCa is expressed at the protein levels indicating this would be a loss-of-function mutation. The authors have only examined expression at the level of mRNA from patients' tumors. The phosphorylation status and protein levels of the endogenous mutant from the tumor cells from the patients also needs to be examined and included in this manuscript.

As the inactive D463N mutant, the D463H protein has a shorter half-life (figure 2a), but not a decreased phosphorylation: on the contrary (see above) we observed an increased phosphorylation at the turn motif and to a lesser extent at the activation loop and hydrophobic motif (figure 3a of the manuscript). This shorter half-life is probably compensated by a higher transcription of the mutant gene (fig 1d, suppl figure 6).

We used IHC to stain chordoid glioma for PKC-alpha as the very limited amount of available human tumor tissues prevented us from performing WB analysis. Furthermore, we note that, should we have sufficient tissue available for WB, we would still be unable to discriminate between the wt and the D463H mutant protein as we lack a D463H PKC-alpha mutant-specific antibody. We need to point out that we have aggressively tried to generate such an antibody but our multiple attempts have so far been unsuccessful.

4. In addition, if cells harboring the mutant were available depletion of the D463N to show decreased proliferation would be more convincing than an overexpression experiment. Correcting the D463N mutant back to WT (via CRISPR/CAS) to show a loss of tumorigenic phenotypes would be the most convincing and compelling approach.

We presume that the reviewer is referring to D463H (= the mutation we discovered here) and not the inactive D463N mutant reported by Antal et al? This would have certainly been a very important experiment but unfortunately we have no such mutant cells from chordoid gliomas as there are no primary or established in vitro cultures, and no in vivo models

possibly because of the slow growing features and exceeding rarity of these tumors. We note that, during the last two years, only one case of chordoid glioma underwent surgical resection at our Center. We tried to culture tumor cells in vitro and establish xenografts in nude mice from this case but both attempts were unsuccessful.

Minor comments:

1. The results that the mutant D463H is localized to cytosol are not novel and expected given the results from the previous study (Antal CE et al, 2014). The authors results should be conveyed in light of the results from this previous publication.

We thank the reviewer for this point, and we mentioned this reference on the revised version of the manuscript.

2. A time course experiment should be conducted to assess if the D463H mutant is less stable than WT (ie this could just be an overexpression experiment examining expression levels at 24, 48, and 72 hours to compare relative levels of expression of WT vs D463H mutant and assess if the D463H is less stable).

We thank this reviewer for this suggestion. We performed protein synthesis block with cycloheximide and showed that the mutant has a much shorter half-life compared to the wild type. This is reported in the new version (figure 2a).

3. It is surprising that Myc-tagged D463H is migrating at a slower rate than WT (Figure 3A)? It suggests this mutant has a higher level of phosphorylation, but the WT should also be migrating at a slower rate unless it has an inactivating mutation? Also WT overexpression will generally result in an increase in MARCKS phosphorylation?? It is almost as if these two lanes were mixed up? Based on the previous report, the D463H should also migrate as a doublet.

Our data show indeed that D463H has a slightly higher level of phosphorylation (figure 3a in the revised version).

4. Traditional semi-quantitative RT-PCR and sanger sequencing should be used to verify increased expression of the D463H compared to WT at the mRNA level (Figure 1D). When comparing expression of the mutant vs WT were both bands utilized to demonstrate the mutant is expressed at higher mRNA levels? Combining both bands would seem to indicate equal levels of the WT and mutant?

We analyzed the band intensity by microfluidics capillary electrophoresis and found that mutant transcripts were indeed expressed at a higher level (approximately x2). This is now more clearly documented on the revised version of the manuscript (Supp Figure 6 and Table S10). We also analyzed the RNAseq and the Sanger data, which found no differences of allele intensity, however the quantification obtained by capillary electrophoresis is probably more sensitive. In addition, since tumor purity is only around 25% (see table S1), this ratio is still probably largely underestimated.

5. PKC is a member of the AGC family of kinases – line 96.

This has been corrected p 5.

Reviewers' comments:

Reviewer #2 (Remarks to the Author):

All my concerns were adequately addressed and I think the manuscript is worth publishing in the current version.

Editorial note

Reviewer 4 commented for the editors only:

This reviewer is not convinced you have provided sufficient evidence supporting the conclusion that the mutation DD463H results in an active protein. In particular the reviewer points out that the identified mutation is positioned in the HRD motif and mutations of the Asp in these regions almost always result in a kinase dead protein. The reviewer suggests that all biochemical data with D463N, D463A, K368M, K368A and N468A should be compared. Reviewer 4 also suggests that in vitro kinase assays with purified proteins would be required as suggested by reviewer 3 and suggests that the transfection based approach used does not allow to support the main conclusion that the kinase is active. Reviewer 4 points out that the use of staurosporine is not suitable as this is not a specific inhibitor. There are some loading controls required (specifically in figure 3) and some literature has not been cited (such as Antal (2015) Cell 160:489-502; McSkimming et al., (2016) Mol Biosystems 12:3651-3665).

Editorial note:

The manuscript was revised accordingly to Reviewer 3 and 4 comments with removal of data referring to an active kinase and toning down of the conclusions where required.